# Honey bees (*Apis mellifera*) modify plant-pollinator network structure, but do not alter wild species' interactions

Sydney H. Worthy ¤*, John H. Acorn, Carol M. Frost

Department of Renewable Resources, University of Alberta, Edmonton, Alberta, Canada

¤ Current address: Parks Department, City of Saskatoon, Saskatoon, Canada
* worthy@ualberta.ca

## Abstract

Honey bees (*Apis mellifera*) are widely used for honey production and crop pollination, raising concern for wild pollinators, as honey bees may compete with wild pollinators for floral resources. The first sign of competition, before changes appear in wild pollinator abundance or diversity, may be changes to wild pollinator interactions with plants. Such changes for a community can be measured by looking at changes to metrics of resource use overlap in plant-pollinator interaction networks. Studies of honey bee effects on plant-pollinator networks have usually not distinguished whether honey bees alter wild pollinator interactions, or if they merely alter total network structure by adding their own interactions. To test this question, we experimentally introduced honey bees to a Canadian grassland and measured plant-pollinator interactions at varying distances from the introduced hives. We found that honey bees increased the network metrics of pollinator and plant functional complementarity and decreased interaction evenness. However, in networks constructed from just wild pollinator interactions, honey bee abundance did not affect any of the metrics calculated. Thus, all network structural changes to the full network (including honey bee interactions) were due only to honey bee-plant interactions, and not to honey bees causing changes in wild pollinator-plant interactions. Given widespread and increasing use of honey bees, it is important to establish whether they affect wild pollinator communities. Our results suggest that honey bees did not alter wild pollinator foraging patterns in this system, even in a year that was drier than the 20-year average.

## Introduction

With the widespread use of managed honey bees (*Apis mellifera*, Linnaeus, 1758) for crop pollination and honey production in parts of the world outside their native range in Europe, Africa, and the Middle East [1], many studies have suggested that high honey bee densities negatively affect wild pollinators through interference or exploitative competition (reviewed in [2, 3]). Wild insect pollinators include many species, including other bees, wasps, and many other orders, including flies, butterflies, moths and beetles, which are diverse in size, sociality,

**Data Availability Statement:** Data available in Dryad: https://doi.org/10.5061/dryad.1vhhmgqzg. Voucher specimens of all insects have been deposited in the E.H. Strickland Entomological Museum, University of Alberta, Edmonton, Canada,

with accession numbers UASM430001-
UASM430601.

**Funding:** S.H.W. was provided funding by the
University of Alberta's Rangeland Research
Institute and awarded funding by the Alberta
Conservation Association. There were no award or
grant numbers associated with these grants. RRI -
https://rri.ualberta.ca/ ACA - https://www.ab-
conservation.com/ The funders had no role in
study design, data collection and analysis, decision
to publish, or preparation of the manuscript.

**Competing interests:** The authors have declared
that no competing interests exist.

and floral trait preferences [4]. Honey bees visit a wide variety of flower species [5], and high
honey bee densities have been linked to declines in wild pollinator diversity [6], abundance
[7], floral resource use [8], and fitness [9], all via competition [10].

However, the first sign of competition between honey bees and wild pollinators may be
changes to which flower species wild pollinators visit in the presence of honey bees, versus
when honey bees are not present [7, 8]. If a wild pollinator is displaced by honey bees from
one or more of its usual flower species (as reported for other competitive relationships between
pollinators; e.g., [11, 12], and for honey bees, [9], the total set of flowers that the wild pollinator
visits will either decrease (as in [13]), or will shift to include new flower species not previously
visited by that pollinator species [14]. Additionally, if honey bee use of a flower depletes nectar
or pollen [15], the frequency of visits by wild pollinator species to that flower species may
decrease, even if visitation does not cease altogether [16]. The ecological effects of these
changes may cascade across the plant-pollinator community, in that a wild pollinator species
displaced by honey bees onto a new flower species may in turn displace the pollinators that
had previously focused on that flower species [17]. Furthermore, when honey bees dominate
floral resources [9], this may also affect a plant's ability to attract specific pollinators and ulti-
mately affect plant fitness and persistence [18]. Thus, shifts in resource overlap patterns in the
plant-pollinator community may signal whether honey bees could cause eventual wild pollina-
tor (or plant) diversity losses. To examine this, an interaction network approach, considering
interaction patterns across the whole community, can be useful [19].

Plant-pollinator interactions can be visualized as bipartite (two trophic-level) networks (Fig
1A), which comprise the set of interactions ("links") observed between species ("nodes").
These networks represent the pattern of interactions between all species within the plant-polli-
nator community that are included in the network model [20]. Honey bees may change the
structure and function of a plant-pollinator network in two different ways: (i) by adding new
interactions (their own interactions with plants), thereby changing network structure without
necessarily indicating competitive displacement [21]; and/or by (ii) altering pre-existing plant-
pollinator interactions, signaling competitive displacement. Changes to network structure
caused by honey bees can be detected by comparing the values for network structural metrics
under different honey bee densities (e.g., [19]). To detect changes to the entire network caused
by honey bees (i), total networks including honey bee interactions must be examined (e.g., [22,
23]; Fig 1A.i–1a.iii). To detect changes in just the set of wild species interactions (ii), networks
not including honey bee interactions must be examined at high versus low honey bee densities
[22, 23; Fig 1A.iv]. This is because network metrics are heavily influenced by the most abun-
dant species [24] which honey bees often are [19]. Therefore, changes to the pattern of wild
species interactions cannot be detected when honey bee interactions are included in the net-
work model. Here we assess both types of networks, while previous studies have typically
tested only total networks (i) [19, 25, 26], but see [27–29]. Importantly, excluding honey bees
from a network model (ii) does not exclude honey bees' ecological effects, because the interac-
tions being visualized in the network occurred when honey bees were present (ii). This is sim-
ply a method of building a network model to clearly see wild pollinator interactions, as in [27–
29] (S1 File).

We used six commonly employed metrics to quantify patterns of resource overlap in plant-
pollinator communities [31], from both the pollinator perspective (how pollinator species
overlap in what plant species they visit), and the plant perspective (how plant species overlap
in what pollinator species visit them). The first metrics provide information on resource over-
lap from the pollinator perspective. The simplest is "generality", which is the mean number of
plant species visited by each pollinator species [31]. We also calculated pollinator "niche over-
lap" [31], which is Horn's information-theory-based index of overlap, ranging from 0 when all

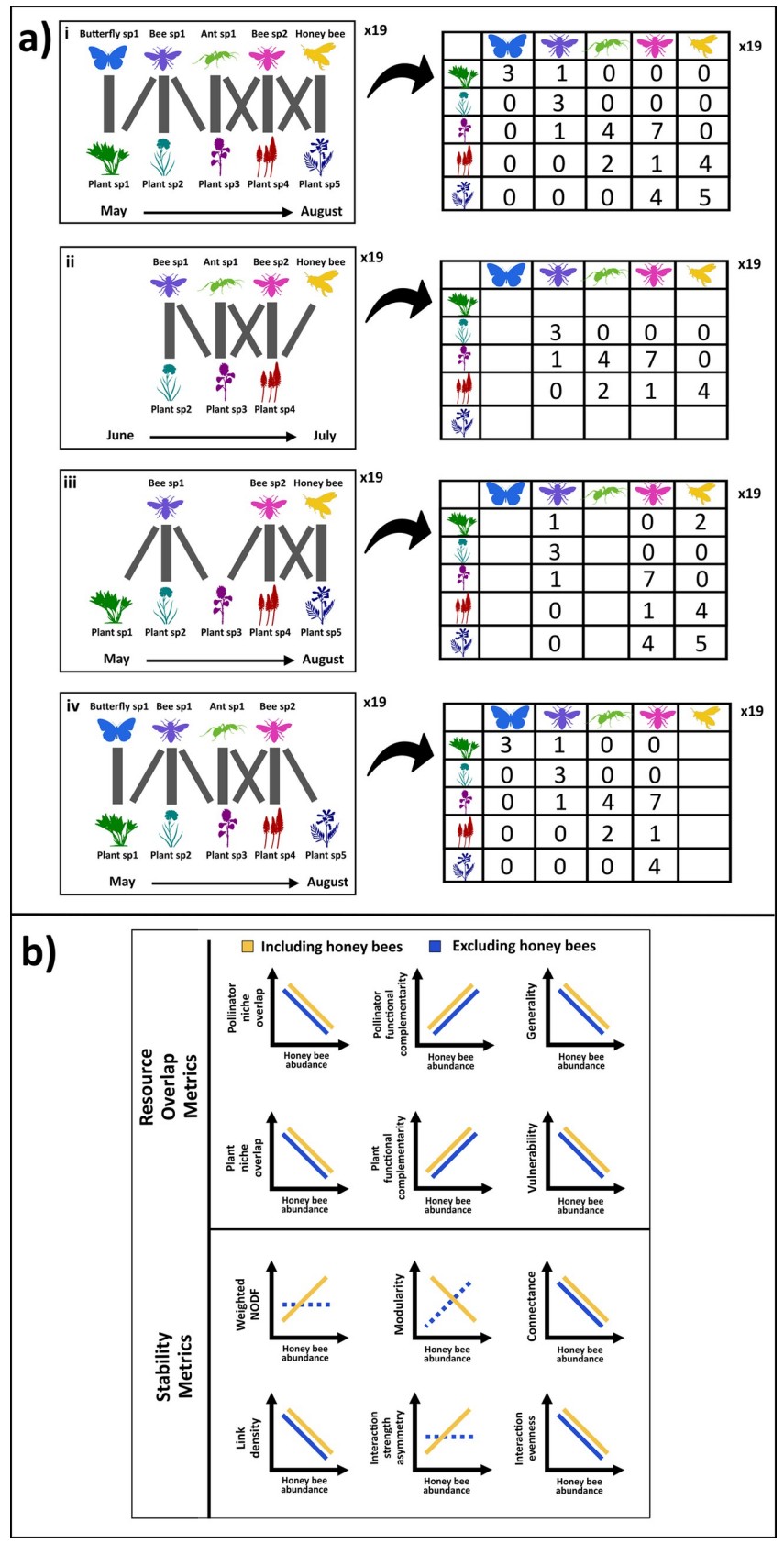

**Fig 1.** Visual representation of the plant-pollinator networks tested (a) and the hypothesized response of each network metric against honey bee abundance (b). a) Plant-pollinator interaction data from each of 19 sampling transects (left), four datasets were created (right): i) a full season all taxa dataset; ii) a mid-season all taxa dataset; iii) a full season bees-only dataset, where only bee interactions, including honey bee interactions, were kept from the full season network; and iv) a without-honey bees dataset, where only honey bee interactions were removed from the full season network. b) Predictions of the effects of increasing honey bee abundance on each network metric, for six metrics of resource overlap and six metrics associated with network stability. Yellow trend lines show predictions for metrics of total network structure, whereas blue trend lines show predictions for metrics calculated from just wild pollinator-plant interactions, excluding honey bee interactions. The rationale and references to support each prediction are in the Supplementary Information. Insect images are from SVG SILH [30].

species' interactions within a trophic level are distinct, to 1 when they are identical [32]. Finally, we calculated pollinator "functional complementarity", a multivariate measure of the dissimilarity of resource use among pollinators. It calculates Euclidean distances between species based on the identity and frequency of their interactions, then clusters the distance matrix to produce a dendrogram, and calculates total branch length of the dendrogram, which provides a univariate expression of this multivariate difference in the set of plant species that each pollinator species interacts with [33]. From the plant perspective, we calculated the equivalent three metrics: "vulnerability" (the mean number of pollinator species visiting each plant species; the name of this metric is from food webs and does not actually refer to vulnerability in plant-pollinator networks), plant "niche overlap" (the Horn's information theoretic measure of overlap in pollinator species visiting each plant species), and plant "functional complementarity" (the index of multivariate difference in the set of pollinator species visiting each plant species).

We hypothesized that honey bees would compete with wild pollinators [10], excluding some wild pollinator species from some of their normal flower species [9], and thus narrowing their floral niches [7]. We therefore predicted that in networks restricted to wild plant-pollinator interactions, we would see decreases in generality and pollinator niche overlap, and an increase in pollinator functional complementarity, with increasing honey bee abundance (Fig 1B). We also hypothesized that honey bees would dominate interactions with their preferred plant species [8], causing these plant species to be visited by fewer pollinator species [27]. We thus predicted that in networks of just wild plant-pollinator interactions, we would see decreases in vulnerability and plant niche overlap and an increase in plant functional complementarity, with increasing honey bee abundance (Fig 1B).

We also calculated six other commonly reported network metrics to maximize comparability of our results with other studies reporting effects of honey bees on plant-pollinator networks [19, 26–28]. These metrics were: nestedness, modularity, connectance, link density, interaction strength asymmetry, and interaction evenness. These additional metrics have been theoretically linked to network stability [34, 35], but given that it is harder to interpret how changes in these metrics relate specifically to competition, we use them mostly as indicators of network structural change, and restrict our discussion of these metrics to the Supplementary Information. Our predictions for the effects of honey bees on the set of stability-related network metrics are shown in Fig 1B, and explained in detail in the Supplementary Information.

We expected to see the strongest effects of honey bees on native plant-pollinator interactions in midsummer (Fig 1A.ii) and therefore analyzed a midsummer dataset separately from the full season dataset. Honey bee colonies comprise between 10,000 to 50,000 adult individuals, attaining their maximum size [36] and foraging distances [37] in midsummer. Larval numbers peak just before adults, demanding greater pollen collection [38].

We also examined how honey bee abundance affected network metrics for a bees-only version of the full season network (Fig 1A.iii). As most studies reporting effects of honey bees on

wild pollinators have only assessed wild bees [3], we wondered if including all pollinating taxa (Hymenoptera, Diptera, Lepidoptera, Hemiptera, Coleoptera) might mask effects on just the wild bee-plant interactions.

Plant-pollinator interaction network studies are complicated by the effects of flower abundance and diversity, which vary spatially (in that pollinators will travel greatly varying distances for plant resources) and temporally (in that different plant species flower at different times of the year), and are not always controlled for. Network metrics also depend on sampling completeness (the proportion of the true interactions that were actually detected; [39], which is not always estimated, making comparability between studies low. Our study controls for effects of all of flower diversity, flower abundance, and sampling effort, and tests experimentally whether changes to plant-pollinator network structure are caused by honey bees changing wild plant-pollinator interactions. Our objectives were to: i) determine if honey bees affect network metrics related to resource overlap and network stability; ii) determine if the effect of honey bees on network metrics is more detectable with higher honey bee colony size; and iii) determine if any effect of honey bees on plant-pollinator network metrics is due to changes in wild plant-pollinator interactions, or due to honey bee-plant interactions.

## Materials and methods

### Experimental design

In 2019, we arranged three clusters of honey bee hives, set at least 3 km apart, on the Mattheis Research Ranch, which is in mixedgrass prairie rangeland near the town of Brooks, Alberta, Canada (Fig 2; GPS locations in S1 Table). Livestock have grazed these rangelands for a century, and this ranch has used fluctuating but light cattle stocking rates (approximately 0.7 animal unit months ha$^{-1}$) since 1970 [40]. This region has average high and low temperatures of 18.3˚C in July and -12.9˚C in January, respectively, and 300 mm of rainfall annually. Honey bees have been present in southern Alberta for over a century, with hive numbers in Alberta increasing by about 4000 hives per year over the past 40 years [41]. Though sporadic bee keeping has occurred in and around the study area in the past, the Eastern Irrigation District, owner of the land surrounding the Mattheis Research Ranch, reported no known apiaries within 19 km of the ranch's boundary line within the study year or the previous year. Apiaries are required by law to register their hive(s) annually to the Provincial Apiculturist. The hive clusters contained 48, 32, and 16 hives respectively, emulating the range of honey bee densities around commercial apiaries. We established eighteen 30 x 2 m transects at 100 m, 500 m, and 5000 m distances on either side of each cluster of honey bee hives for a total of six replicates (Fig 2). The 100 m and 500 m distances were chosen to achieve high and medium honey bee densities. We chose 5000 m as the minimum distance for sites intended to be without honey bees, because on average they do not tend to travel farther than this distance [42].

Between May 28 and August 28, 2019, observers visited each transect and surveyed insect flower visitation almost once per week (weather permitting), for a total of 10 collection rounds. The first transect sampled at the beginning of each day was chosen on a rotating schedule, and subsequent transects were sampled along a route of highest efficiency within that day. During some collection rounds, some transects could not be sampled due to cattle presence or a lack of flowers, resulting in different amounts of sampling for each transect (S1 Table). Transects were visited only on warm ($\geq$ 15˚C), sunny days with winds under 50 km/h [45, 46]. We measured wind speed with a Brunton Sherpa (Riverton, Wy, USA). Sampling occurred between 9:30 am and 5:00 pm, when insect flower visits are most frequent [47].

Transects were observed for 30 minutes by two people for a total of 60 person-minutes per transect per collection round (4200 total collection minutes). When large volumes of insects

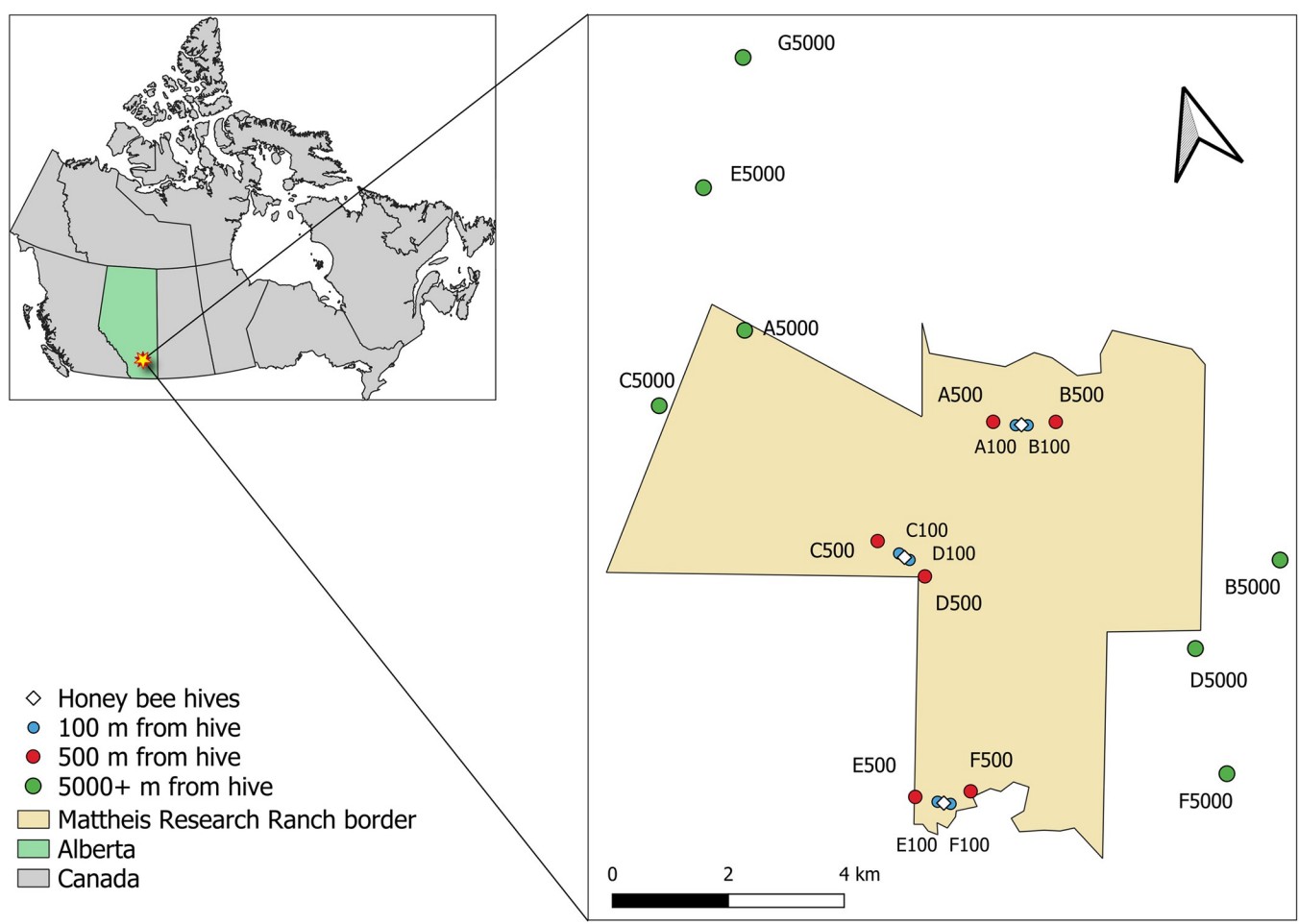

**Fig 2. The research location, Mattheis Ranch, in southern Alberta, Canada.** Sampling transects are shown as circles and honey bee hives as diamonds. Each apiary contained a cluster of hives (northernmost: 48 hives, central: 32 hives, southernmost: 16 hives). The transect G5000 indicates the new location established when sampling was discontinued at F5000 after July 9th, 2019. Land outside the Mattheis Ranch is managed by the Eastern Irrigation District. Map shapefiles from Natural Resources Canada under an Open Government license [43]. Map created in QGIS [44].

were captured, sampling was paused via a timer for processing and then resumed. All insects that visibly contacted the anthers or stigma of open flowers were collected with a hand net and placed in labelled individual vials, frozen, and identified to species in the lab. Some of these specimens (5%) were identified to morphospecies. When insects observed landing on flowers were missed they were recorded, but these data were excluded because the species-level identifications for these observations were heavily biased toward larger-bodied pollinators. All identifications were completed using a dissecting microscope (Bausch & Lomb Stereozoom, Wild M8, or Leica MZ 125), dichotomous keys, expert taxonomist help, and comparison with reference specimens from the University of Alberta Strickland Museum and University of Calgary Museum of Zoology entomological collections. Following flower visitor sampling, all flowering plants on the transect were identified and their flowers were counted. A list of all references used for the insect and plant species identifications is included in S2 Table. Occasionally, when flowers were not present on the marked transect, we moved the transect ≤ 10 m from its original location to reach flowers near the original transect demarcation. Moving the transect did not change the distance to the hives.

After July 9, 2019, sampling at one 5000 m transect "F5000" was halted after many honey bees were observed (presumably from a feral or unregistered hive, see S1 Fig), since this replicate was intended to be a low-honey bee density transect. A new 5000 m transect, "G5000", was selected approximately 8000 m away from the northern 48-hive cluster (Fig 2). We treated these transects as distinct site replicates but controlled statistically for the lower number of collections at these by including number of collections as a covariate in our multiple regression models (see below and S1 Table).

This study involved the collection of plant and animal materials from the Mattheis Research Ranch, a research area owned by the University of Alberta, and the Eastern Irrigation District (EID) in Southern Alberta. Permission was granted by both and did not require a field permit number.

## Network metric calculation

We pooled the observed insect-flower visitor interactions across the season for each transect, to construct one bipartite network for each transect, for a total of 19 networks (6 x 3 distances from hives, plus the new 5000 m transect "G5000" that was established mid-season).

We calculated the six network metrics described above to analyze honey bee effects on resource use overlap: generality, vulnerability, pollinator niche overlap, plant niche overlap, pollinator functional complementarity, and plant functional complementarity [31]. "Weighted" versions were used, meaning that interactions were weighted by their frequency, rather than just considering interaction presence or absence in calculating the metrics [48].

Additionally, we calculated the weighted versions of six network metrics to analyze the effects of honey bee abundance on network structural metrics related to stability: WNODF, a nestedness metric based on overlap and decreasing fill ("nestedness") [49], modularity, interaction strength asymmetry, interaction evenness, connectance, and link density. Calculation of these metrics and their interpretation is described in detail in the Supplementary Information. Each metric was calculated using the bipartite package [50]. All statistical analyses were completed using R version 3.2.4. [51]. Additionally, for comparability to [26], in a separate analysis, we pooled the network interactions across the season for the 100 m, 500 m, and 5000 m transects into meta-networks, and calculated each aforementioned network metric for each meta-network and qualitatively compared them in the results.

## Statistical analysis

We used linear regression to test the effect of honey bee abundance on each of the six resource-overlap-related and the six stability-related network metrics. We assessed Pearson correlation coefficients between these metrics and found pollinator and plant functional complementarity were highly correlated, as well as link density and vulnerability (S3 Table). We kept all metrics in the analysis, because these were ecological results rather than consequences of the metric definitions and might differ for different datasets. We pooled honey bee abundance (total number of honey bees observed visiting flowers) across the full season for each transect, and divided by number of collection rounds at that transect, to calculate "honey bee abundance" (S1 Fig), which we used as the predictor variable in all full season statistical models. We used honey bee abundance rather than treatment (distance from honey bee hives) as the predictor variable because distance from hives was not as closely associated with honey bee abundance as might have been expected (S1 Fig). Honey bee abundance seemed like a better representation of honey bee "effect" at that transect, as in [25, 28].

We tested the effect of honey bee abundance on each network metric for several datasets: i) each transect's network from the full season ("Full season all taxa"; Fig 1A.i); ii) each transect's

network for only the mid-season sampling rounds, using honey bee abundance pooled across the mid-season divided by number of mid-season collection rounds at that transect as the predictor variable in this case ("Mid-season all taxa"; Fig 1A.ii described below); and iii) only bee interactions from the full season network ("Full season bees-only"; Fig 1A.iii). In these analyses each network included the observed honey bee interactions, as well as the wild pollinator interactions.

We first ran a simple linear regression (SLR) with honey bee abundance as the only predictor variable for each response variable (network metric) in each dataset. Some of the transects (replicates) were close together, so we plotted the standardized residuals of each SLR to visualize whether similar residuals were close together in space, and to examine whether a special correlation structure was necessary to account for any spatial autocorrelation (S2 Fig).

We then ran generalized least squares (GLS) models with different correlation structures (no correlation, corEcp, corGaus, corSpher, corLin, corRatio) for each response variable (nlme package, [52]). After all models with correlation structures (including no special correlation structure) were run, the model with the lowest AICc value was selected (MuMin package, [53]). The best models for each predictor variable did not include special correlation structures, so simple linear regression models (SLR) were run. The exception was for modularity in the without-honey bees network, so for this metric in this dataset, we ran a GLS model with the corRatio correlation structure.

We then ran an additional multiple regression model (MR) for each response variable in each dataset for which honey bee abundance was significant in the SLR. This was because flower abundance, number of available flower species, and collection effort were positively correlated with honey bee abundance (S4 Table), so we included them as additional predictor variables in this second set of models, to test for an effect of honey bees above and beyond any correlation of honey bee abundance with these other variables [54]. The predictor variables included in each full MR model were honey bee abundance, flower abundance, flower species richness, total number of collection rounds, the interaction between honey bee abundance and flower abundance, as well as the interaction between honey bee abundance and flower species richness. All continuous predictor variables were standardized by subtracting each transect's value from the mean and dividing by the standard deviation. Then, the full model and all possible simpler models were run, for a total of 21 regression models per response variable (nlme package, [52]), and the best model for each response variable was selected using AICc. In this second set of models, if honey bee abundance was retained after model selection and was significant, it would suggest that there was an effect of honey bees on that response variable that could not plausibly be attributed to other correlated predictor variables [54]; Supplementary Information). If honey bee abundance was significant in the SLR model, but was either not retained or not significant for the MR model, it would suggest that response variable is related to honey bee abundance, but any effects of honey bees cannot be unambiguously attributed to honey bees. They may be due instead to flower abundance or diversity or collection effort, which honey bees were responding to themselves.

The assumptions of normality, homogeneity of variance, and linearity for each response variable were tested. If any assumptions were not met, that response variable was log transformed and model selection was repeated, after which, if assumptions were still not met, a Box-Cox transformation was applied (MASS package, [55]). Transformation did not improve normality or homogeneity of variances for the model for bees-only interaction strength asymmetry, so the results of this model were not interpreted as we could not find a robust modeling method.

To test whether the effects of honey bee abundance on the network metrics differed mid-season, we created a mid-season all taxa dataset (Fig 1A.ii). To determine which sampling dates the mid-season dataset should include, we divided the total honey bee abundance across the season, from all transects pooled, into three roughly equal-length periods: "early", "mid"

and "late", based on natural breaks in the abundances of honey bees (S3 Fig). We then tested the effects of honey bee abundance on all the same response variables as above, using the procedure outlined above, for this mid-season all taxa dataset. Because some transects' networks were too species-poor in the mid-season to obtain accurate values (A100, B100, B500, E100, F5000), they were removed from this analysis.

Finally, we created a reduced network from the full season all taxa dataset where all honey bees were removed from the network, to look at the network metrics for wild pollinators only (Fig 1A.iv; [28]). In this "without-honey bees" dataset we ran SLR and MR models that tested the effect of honey bee abundance and all predictor variables, respectively, on all the same response variables. If honey bee abundance was a significant predictor in these without-honey bees models, it would indicate that honey bees altered how wild pollinators interacted; if it was not, it would indicate that the changes to that network metric caused by honey bee abundance in models for our other datasets were just due to the honey bee node and links contributing to the network structure.

This analysis involved interpreting 65 separate models, so to maintain Type I Error at 0.05, a Bonferroni-Holm correction was used in assessing significance of the final models (at $P = 0.00077$). However, all $P$-values are presented to allow a less conservative consideration of uncorrected $P$-values. To estimate the interaction sampling completeness, we divided the raw interaction richness across all transects by the Chao1 estimated total interaction richness across all transects (SpadeR, [56]). This calculated the proportion of the estimated total number of unique interactions that our sampling detected (as in [39]). We repeated this at the transect level for the full season all taxa, mid-season all taxa, and full season bees-only datasets.

## Results

### Diversity of pollinators, plants, and interactions in the grassland plant-pollinator community

A total of 281 pollinator species and 37 plant species were identified in 1,814 interactions, with 654 unique species pair interactions. Of these interactions, 425 were only observed once. The Chao1 estimated true total number of unique interactions was 1,779 interactions (95% confidence interval: 1500–2148 interactions), meaning that we observed 31–44% of the estimated total number of plant-pollinator interactions occurring in this study region (S4 Fig). We also calculated the Chao1 estimated true number of unique interactions at each transect, for the full season, mid-season, and full season bees-only datasets (S5 Table). At 100 m transects, honey bees accounted for 29.4% of the total interactions, at 500 m transects, 14.9% of the total interactions, and at 5000 m transects, 1% of the total interactions. Bees (including honey bees) accounted for 872 (48.1%) of the total interactions. Of the remaining interactions, flies accounted for 494 (27.2%), butterflies for 130 (7.2%), ants for 104 (5.7%), beetles for 100 (5.5%), wasps for 62 (3.4%), true bugs for 27 (1.5%), and day-flying moths for 25 (1.4%). In the mid-season all taxa dataset, at 100 m transects, honey bees accounted for 49.6% of the interactions, at 500 m transects, 25.2% of the interactions, and at 5000 m transects, 2.4% of the interactions. The meta-networks of pooled interactions for each transect are shown in Fig 3. All species identifications are listed in S6 (pollinators) and S7 (plants) Tables.

### Effect of honey bees on the structure of networks including honey bee interactions

In the full season all taxa dataset, in the SLR models (testing honey bee abundance as the only predictor against each network metric), increasing honey bee abundance was associated with

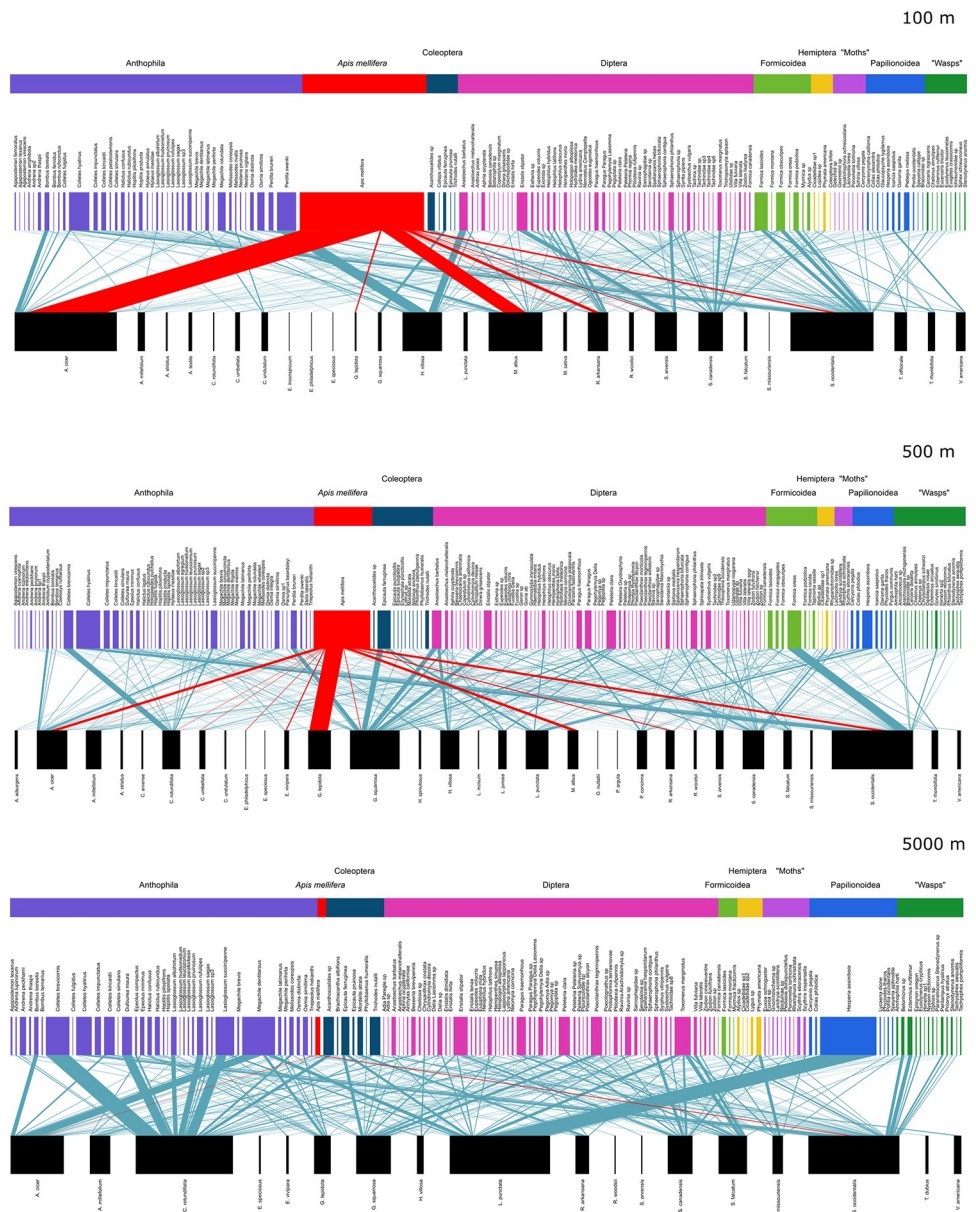

**Fig 3. Plant-pollinator meta-networks from the full season all taxa dataset.** Data from transects 100 m (top), 500 m (middle) and 5000 m (bottom) from honey bee hives are pooled across the season. The upper row depicts the pollinator species, coloured by their given taxonomic "group" or Order (see S6 Table for the list of pollinator species). "Moths" are Lepidoptera excluding butterflies (Papilionoidea). "Wasps" are Aculeata excluding ants (Formicoidea) and bees (Anthophila). The bottom row (in black) depicts the plant species (see S7 Table for the list of plant species). The width of each upper and lower bar represents the relative frequency of interactions observed for that species. The interactions between plants and wild pollinators are represented by blue lines; width indicates the frequency of the interaction. Honey bees and their interactions are indicated in red.

significant changes in two network metrics related to resource overlap. Pollinator functional complementarity ($t = 5.92$, $P < 0.0001$) and plant functional complementarity ($t = 6.39$, $P < 0.0001$) significantly increased with honey bee abundance, indicating decreasing similarity in resource use between species within a trophic level with increasing honey bee abundance (Fig 4A, Table 1A). Increasing honey bee abundance was also associated with a significant

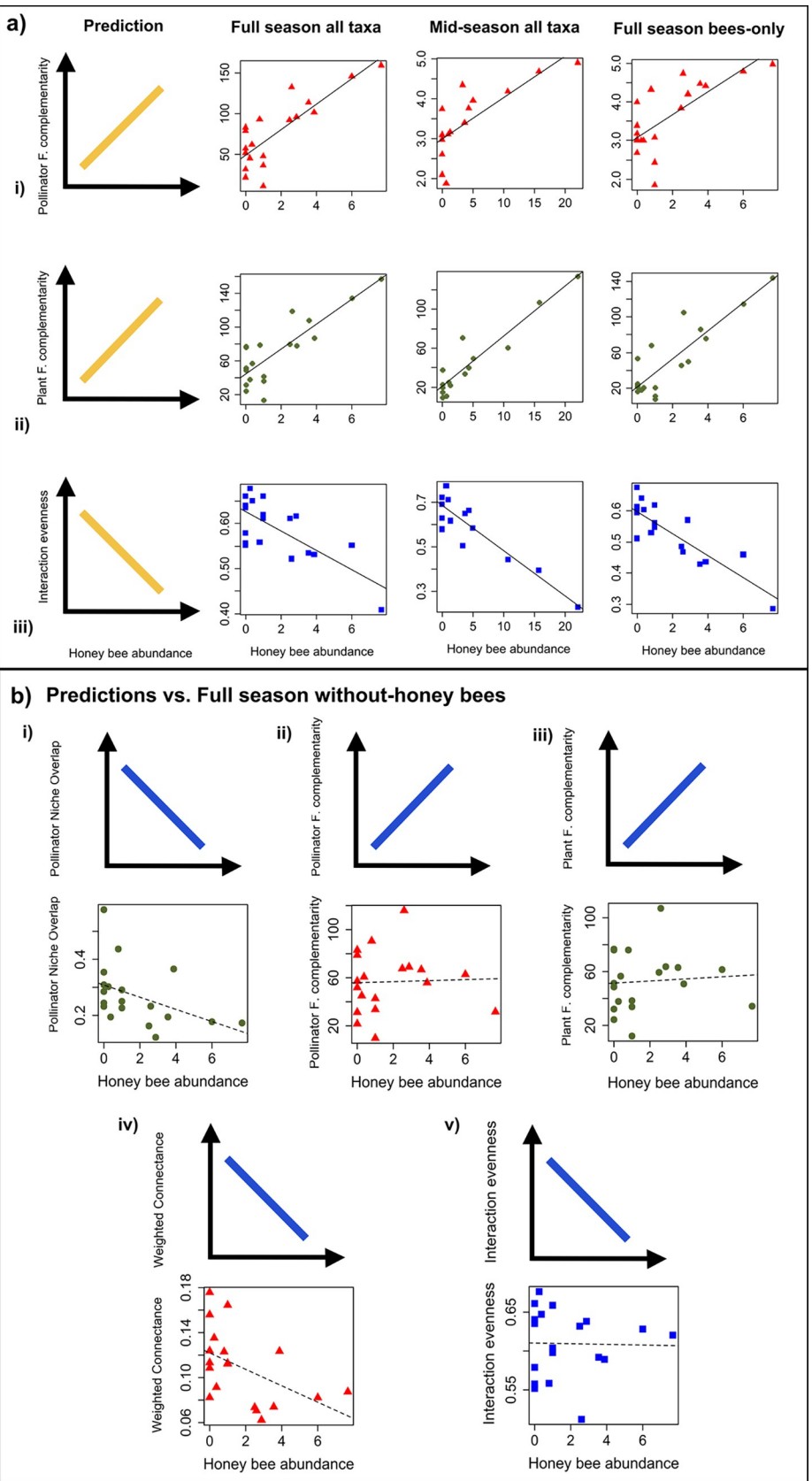

**Fig 4. Predictions vs. results of network metric analyses from with-honey bees networks.** (a) and without-honey bees networks (b). a) Relationships between honey bee abundance and the network metrics: (i) pollinator functional complementarity, (ii) plant functional complementarity, and (iii) interaction evenness that were significantly related to honey bee abundance across three datasets (full season all taxa, mid-season all taxa, and full season bees-only). Solid regression lines indicate significant relationships with Bonferroni-Holm correction, whereas dotted lines indicate insignificant relationships. b) Relationships between honey bee abundance and the network metrics: (i) pollinator niche overlap, (ii) pollinator functional complementarity, (iii) plant functional complementarity, (iv) weighted connectance, and (v) interaction evenness in the without-honey bees dataset. Dashed regression lines indicate non-significant relationships with Bonferroni-Holm correction. Although (i) and (iv) appear to contain trends, these cannot be unambiguously attributed to honey bee abundance, as they can be explained by flower community variables and collection effort (Table 2D) that were somewhat correlated with honey bee abundance (S4 Table).

decrease in interaction evenness ($t$ = -4.33, $P$ = 0.0005; Fig 4A, Table 1A). For the mid-season all taxa dataset (Fig 4A, Table 1B), and the full season bees-only dataset (Fig 4A, Table 1C), the results were very similar. All reported $P$-values are significant after the Bonferroni-Holm correction.

In the second set of MR models (including all predictor variables), flower species richness and total number of collections best explained most response variables (Table 2A). However, increasing honey bee abundance was still associated with significant increases in pollinator functional complementarity ($t$ = 5.51, $P$ < 0.0001) and plant functional complementarity ($t$ = 5.75, $P$ < 0.0001), and a significant decrease in interaction evenness ($t$ = -4.33, $P$ = 0.0005) (Table 2A), indicating that these effects on plant-pollinator network structure could be attributed to honey bees, rather than to flower community variables that were correlated with honey bee abundance. For the mid-season all taxa dataset (Table 2B) and the full season bees-only dataset (Table 2C), the results were very similar, except that in the full season bees-only dataset, pollinator functional complementarity was not significantly increased by honey bees after the Bonferroni-Holm correction, even though it still had a very low P-value (t = 3.57, P = 0.0026).

### Effect of honey bees on the without-honey bees network structure

In contrast to all of the above results, the full season without-honey bees dataset with only honey bee abundance as a predictor variable (SLR models) showed that honey bees did not significantly affect pollinator functional complementarity, plant complementarity, or interaction evenness, or any other network metric (Fig 4B, Table 2D). Although some P-values were low in the SLR models (Table 1D), in the MR models, honey bee abundance was removed during model selection in every case (Table 2D), suggesting that the low P-values in the SLR models could be explained by correlated effects of the flower variables or collection effort better than they could be explained by honey bee abundance. This disappearance of any significant effect of honey bee abundance on any network metrics when honey bees were not considered as part of the network was not an artifact of having decreased the network size by removing the honey bee interactions (see comparison of all network sizes in S8 Table).

### Discussion

As predicted, total plant-pollinator network structure (including honey bee interactions), changed with increasing honey bee abundance, as reported in other studies [19, 25, 26, 28]. Across three datasets (full season all taxa, mid-season all taxa, and full season bees-only), the changes due to honey bees (rather than potentially due to flower community variables correlated with honey bee abundance) were that pollinator functional complementarity and plant functional complementarity significantly increased, and interaction evenness significantly decreased as honey bee abundance increased. The effect sizes were highest in the mid-season all taxa dataset, when honey bee abundances were highest (compare regression coefficients in

**Table 1. Results of the simple linear regression (SLR) model with the lowest AICc values for each response variable, where the full model for each response variable contained honey bee abundance as the only predictor variable.**

| Response variable | Confidence intervals (95%) | | Adjusted $R^2$ | Regression coefficient | $t$-value | $P$-value |
|---|---|---|---|---|---|---|
| a) Full season all taxa | | | | | | |
| log(Generality) | -0.06 | 0.11 | -0.05 | 0.0238 | 0.58 | 0.5720 |
| Vulnerability | -2.15 | 0.42 | 0.07 | -0.8657 | -1.44 | 0.1710 |
| log(Pollinator niche overlap) | -0.31 | 0.03 | 0.13 | -0.1398 | -1.82 | 0.0910 |
| Plant niche overlap | -0.02 | 0.01 | -0.05 | -0.0040 | -0.55 | 0.5910 |
| Pollinator functional complementarity | 22.27 | 46.91 | 0.65 | 34.5940 | 5.92 | **<0.0001** |
| Plant functional complementarity | 21.72 | 43.12 | 0.69 | 32.4210 | 6.39 | **<0.0001** |
| Weighted nestedness | -1.59 | 0.92 | -0.05 | -0.3371 | -0.58 | 0.5732 |
| Modularity | -0.05 | 0.05 | -0.06 | 0.0011 | 0.05 | 0.9640 |
| Weighted connectance | -0.04 | -0.01 | 0.46 | -0.0262 | -4.07 | 0.0008 |
| Link density | -1.08 | 0.37 | 0.00 | -0.3514 | -1.02 | 0.3203 |
| Interaction strength asymmetry | -0.17 | -0.01 | 0.18 | -0.0879 | -4.33 | 0.0381 |
| Interaction evenness | -0.07 | -0.02 | 0.50 | -0.0475 | -4.33 | **0.0005** |
| b) Mid-season all taxa | | | | | | |
| Generality | -0.12 | 0.12 | -0.12 | 0.0023 | 0.04 | 0.9660 |
| Vulnerability | -4.71 | 0.13 | 0.30 | -2.2903 | -2.18 | 0.0605 |
| Pollinator niche overlap | -0.19 | 0.08 | -0.02 | -0.0596 | -0.93 | 0.3786 |
| Plant niche overlap | -0.21 | 0.09 | -0.01 | -0.0154 | -0.48 | 0.6474 |
| log(Pollinator functional complementarity) | 0.33 | 0.96 | 0.57 | 0.6423 | 4.41 | **0.0007** |
| Plant functional complementarity | 25.05 | 38.66 | 0.88 | 31.8550 | 10.11 | **<0.0001** |
| Weighted nestedness | -3.45 | 5.85 | -0.08 | 1.2010 | 0.60 | 0.5679 |
| Weighted connectance | -0.07 | 0.00 | 0.22 | -0.0360 | -2.22 | 0.0450 |
| Link density | -1.61 | -0.14 | 0.29 | -0.8758 | -2.58 | 0.0227 |
| Interaction strength asymmetry | -0.21 | 0.04 | 0.09 | -0.0880 | -1.53 | 0.1497 |
| Interaction evenness | -0.16 | 0.96 | 0.57 | -0.1242 | -7.94 | **<0.0001** |
| c) Full season bees-only | | | | | | |
| Generality | -0.09 | 0.21 | -0.02 | 0.0617 | 0.88 | 0.3960 |
| log(Vulnerability) | -0.20 | 0.01 | 0.15 | -0.0927 | -1.86 | 0.0851 |
| Pollinator niche overlap | -0.10 | 0.03 | 0.04 | -0.0375 | -1.28 | 0.2240 |
| Plant niche overlap | -0.06 | 0.05 | -0.08 | -0.0031 | -0.12 | 0.9058 |
| log(Pollinator functional complementarity) | 0.34 | 0.96 | 0.51 | 0.6516 | 4.42 | **0.0004** |
| Plant functional complementarity | 24.96 | 44.42 | 0.76 | 34.6900 | 7.52 | **<0.0001** |
| Weighted nestedness | -2.78 | 2.38 | -0.07 | -0.1993 | -0.17 | 0.8700 |
| Weighted connectance | -0.06 | -0.01 | 0.32 | -0.0361 | -3.05 | 0.0073 |
| Link density | -0.36 | 0.15 | -0.01 | -0.1088 | -0.90 | 0.3820 |
| Interaction strength asymmetry[a] | -0.23 | 0.06 | 0.03 | -0.0376 | -1.24 | 0.2333 |
| Interaction evenness | -0.10 | -0.05 | 0.68 | -0.0777 | -6.33 | **<0.0001** |
| d) Full season without-honey bees | | | | | | |
| log(Generality) | -0.04 | 0.14 | 0.02 | 0.0482 | 1.15 | 0.2660 |
| Vulnerability | -1.12 | 1.66 | -0.05 | 0.2684 | 0.41 | 0.6890 |
| log(Pollinator niche overlap) | -0.35 | -0.02 | 0.21 | -0.1877 | -2.40 | 0.0279 |
| Plant niche overlap | -0.02 | 0.01 | 0.00 | -0.0069 | -0.95 | 0.3530 |
| Pollinator functional complementarity | -12.18 | 14.07 | -0.06 | 0.9445 | 0.15 | 0.8810 |
| Plant functional complementarity | -9.67 | 13.14 | -0.05 | 1.7320 | 0.32 | 0.7530 |
| Weighted nestedness | -1.58 | 0.24 | 0.09 | -0.6727 | -1.58 | 0.1350 |
| Modularity (*CorRatio*)[b] | 0.00 | 0.02 | NA | 0.0081 | 1.25 | 0.2282 |

(*Continued*)

**Table 1.** (Continued)

| Response variable | Confidence intervals (95%) | | Adjusted R² | Regression coefficient | t-value | P-value |
|---|---|---|---|---|---|---|
| Weighted connectance | -0.03 | 0.00 | 0.20 | -0.0162 | -2.34 | 0.0320 |
| Link density | -0.55 | 0.88 | -0.04 | 0.1616 | 0.48 | 0.6390 |
| Interaction strength asymmetry | -0.16 | 0.03 | 0.07 | -0.0669 | -1.52 | 0.1470 |
| Interaction evenness | -0.02 | 0.02 | -0.06 | -0.0009 | -0.09 | 0.9310 |

Results are shown for the three datasets: a) the full season all taxa dataset, b) the mid-season all taxa dataset, and c) the full season bees-only dataset, all of which included the honey bee interactions, and d) the full season without-honey bees dataset. The regression coefficients (slopes) are a measure of the expected change in the response variable associated with a one-unit change in the predictor. Bolded P-values denote significance with Bonferroni-Holm correction (P < 0.00077). Transformations are listed with the response variable.

[a] Assumptions of normality and homogeneity of variance were not met for this statistical test, so P-value is not likely accurate and is not interpreted.

[b] Adjusted R² cannot be calculated for a model fit using generalized least squares.

Table 2A to partial regression coefficients in Table 2B). The exception was that pollinator functional complementarity did not change significantly in the bees-only dataset (Table 2C). When looking at the meta-networks (Fig 3), which had interactions pooled across transects by distance from hives, similar to [26], rather than pooled by transect, the results were qualitatively similar (S9 Table), suggesting that our results are robust across different analysis methods.

However, when considering only wild plant-pollinator interactions in the without-honey bees dataset, honey bee abundance (which varied from high to almost zero between these sites) did not affect any of the network metrics (Table 2D), suggesting that all changes to network structure associated with honey bee abundance in this study were caused by the honey bees' interactions contributing to the network structure, rather than by honey bees altering the interactions of wild pollinators and plants. Thus, adding honey bees had no detectable effect on the network structure of the wild plant-pollinator interactions in this region, whether considering metrics of resource overlap, or metrics associated with network stability.

Magrach et al. [19], who reported changes in plant-pollinator network structure between orange groves in Spain in which honey bees accounted for 72% vs. 38% of interactions, found similar results to our with-honey bee networks: honey bees decreased interaction evenness and increased functional complementarity in their system. The increase in functional complementarity indicates increasing dissimilarity in the interaction composition of pollinators and plants in the network, which can be thought of as a reduction of their resource use overlap [33]. Although this metric accounts for the interactions of all species in the community, its value can change in response to the addition of an abundant generalist species' interactions, such as the honey bee, because an abundant generalist species' interaction identities and frequencies are so different from those of less abundant and generalist species (corresponding to a long dendrogram branch length). However, Magrach et al. [19] did not test with- versus without-honey bee networks for honey bee abundance effects on these metrics. Although their with-honey bee network results are similar to ours, we found that in our without-honey bees network, the effect of honey bee abundance disappeared, suggesting no effect of honey bees on wild plant-pollinator interactions.

Similarly, Valido et al. [26], who tested changes in network structure before vs. after honey bees were added to one site in Tenerife, Canary Islands in three consecutive years, did not remove the honey bee node from the networks and re-test effects of honey bee abundance on just the wild pollinator interactions. Additionally, they did not control for floral abundance or flower species richness as potentially correlated predictor variables. Furthermore, we had

**Table 2. Results of the multiple regression (MR) model with the lowest AICc value for each response variable.**

| Response variable | Predictor variables retained in final model | Confidence intervals (95%) | | Adjusted R$^2$ | Partial regression coefficient | t-value | P-value |
|---|---|---|---|---|---|---|---|
| **a) Full season all taxa** | | | | | | | |
| Pollinator functional complementarity | honey bee | 16.13 | 36.34 | | 26.2380 | 5.51 | **<0.0001** |
| | collection | 8.79 | 29.00 | 0.81 | 18.8970 | 3.97 | 0.0011 |
| Plant functional complementarity | honey bee | 16.33 | 35.39 | | 25.8630, | 5.75 | **<0.0001** |
| | collection | 5.30 | 24.36 | 0.80 | 14.8320 | 3.30 | 0.0045 |
| Weighted connectance | flower species richness | -0.03 | -0.01 | 0.68 | -0.0210 | -3.50 | 0.0030 |
| | collection | -0.03 | 0.00 | | -0.0146 | -2.43 | 0.0271 |
| Interaction strength asymmetry | flower species richness | -0.19 | -0.04 | 0.33 | -0.1136 | -3.25 | 0.0047 |
| Interaction evenness | honey bee | -0.07 | -0.02 | 0.50 | -0.0475 | -4.33 | **0.0005** |
| **b) Mid-season all taxa** | | | | | | | |
| log(Pollinator functional complementarity) | honey bee | 0.51 | 0.97 | | 0.7395 | 6.98 | **<0.0001** |
| | flower species richness | 0.17 | 0.62 | 0.79 | 0.3952 | 3.76 | 0.0027 |
| Plant functional complementarity | honey bee | 28.50 | 39.16 | | 33.8290 | 13.84 | **<0.0001** |
| | flower species richness | 2.74 | 13.32 | 0.93 | 8.0290 | 3.31 | 0.0062 |
| Weighted connectance | honey bee | -0.07 | -0.02 | | -0.0460 | -3.62 | 0.0035 |
| | flower species richness | -0.07 | -0.01 | 0.55 | -0.0409 | -3.23 | 0.0072 |
| Link density | honey bee | -1.61 | -0.14 | 0.29 | -0.8758 | -2.58 | 0.0227 |
| Interaction evenness | honey bee | -0.16 | -0.09 | 0.82 | -0.1242 | -7.94 | **<0.0001** |
| **c) Full season bees-only** | | | | | | | |
| log(Pollinator functional complementarity) | honey bee | 0.18 | 0.72 | | 0.4534 | 3.57 | 0.0026 |
| | collection | 0.18 | 0.72 | 0.71 | 0.4481 | 3.52 | 0.0028 |
| Plant functional complementarity | honey bee | 20.16 | 39.07 | | 29.6150 | 6.64 | **<0.0001** |
| | collection | 2.03 | 20.93 | 0.82 | 11.4760 | 2.57 | 0.0204 |
| Weighted connectance | flower species richness | -0.07 | -0.03 | 0.58 | -0.0476 | -5.20 | **0.0001** |
| Interaction evenness | honey bee | -0.10 | -0.05 | 0.68 | -0.0887 | -4.56 | **0.0003** |
| **d) Full season without-honey bees** | | | | | | | |
| log(Pollinator niche overlap) | flower species richness | -0.11 | -0.03 | 0.39 | -0.0704 | -3.54 | 0.0025 |
| Pollinator functional complementarity | collection | 6.85 | 26.72 | 0.39 | 16.7810 | 3.56 | 0.0024 |
| Plant functional complementarity | collection | 5.33 | 23.02 | 0.37 | 14.1780 | 3.28 | 0.0035 |
| Weighted connectance | flower species richness | -0.03 | -0.00 | | -0.0156 | -2.65 | 0.0174 |
| | collection | -0.03 | -0.00 | 0.61 | -0.0144 | -2.46 | 0.0256 |
| Interaction evenness | flower species richness | -0.01 | 0.03 | 0.00 | -0.0100 | 0.97 | 0.3450 |

The full MR model for each response variable contained honey bee abundance, flower abundance, flower species richness, the interactions between honey bee abundance and flower abundance, and between honey bee abundance and flower species richness, and collection effort as predictor variables. Only response variables for which the P-value in the corresponding simple linear regression (SLR) model was ≤ 0.00077 are shown here, with the exception of d), where they were also included if a) was ≤ 0.05. Results are shown for four datasets: a) the full season all taxa dataset, b) the mid-season all taxa dataset, c) the full season bees-only dataset, all of which included the honey bee interactions, and d) the full season without-honey bees dataset. The partial regression coefficients are the expected change in the response variable associated with a one unit change in a predictor variable holding the other predictor variables constant. Bolded P-values denote significance with Bonferroni-Holm correction (P < 0.00077). Transformations are listed with the response variable.

similar sampling effort, range of honey bee abundances tested, and estimated sampling completeness to both [19] and [26]. Taken together, this suggests that if more potentially correlated variables had been controlled for, these two studies may also have found a non-significant effect of honey bees on wild plant-pollinator network structure. Alternatively, it may be that honey bees shift wild pollinator interactions in other systems more than in the grassland

system where our study took place. Lázaro et al. [28], similar to our study, tested honey bee effects on total network structure, as well as on the network structure of just wild pollinator interactions, and found larger effects on total network structure. They reported that the effects of honey bee abundance on wild bee network metrics that they observed were caused by changes in wild bee species richness, whereas in our study, wild pollinator species richness did not change with increasing honey bee abundance.

The most unexpected result from our study was that we found no effect of honey bees on nestedness, whether honey bee interactions were included or not (Table 1A–1D). This contrasts with several studies that have reported increases in network nestedness when honey bees were present [21–23, 28, 57]. However, all these studies except [28] tested honey bee effects on "binary" nestedness, calculated without considering interaction frequencies, whereas we used "weighted" nestedness, which incorporates interaction frequency, because [58] showed that changes to weighted nestedness are a better indicator of shifts in resource partitioning. Our finding of no change to weighted nestedness associated with increasing honey bee abundance is further evidence that honey bees did not change patterns of resource partitioning by wild pollinators or plants [58]. Our findings for the other stability-related metrics for the with-honey bees networks are similar to the findings of other studies (see Supplementary Information).

Our failure to find evidence that honey bees shift resource use by wild pollinators, or shift visitation of plant species, may indicate that honey bees do not compete with wild pollinators in this system, due to abundant floral resources that are not limiting. Additionally, honey bees interacted frequently with non-native plants. Alberta's grasslands are diverse and dominated by native plant species, but several exotic plants exist in the region; sweet clover (*Melilotus* spp.), sow-thistle (*Sonchus* spp.), and cicer milk-vetch (*Astragalus cicer*), in particular, exist in scattered patches throughout the study region. We observed that these exotic species were highly attractive to pollinators of all types. In their network study, Bendel et al. [59] determined that honey bees preferred exotic plants over native plants. We also observed that small-flowered and small-stemmed plants were less appealing to larger-bodied pollinators. This preference may have prevented honey bees from outcompeting wild pollinators for resources, preserving native plant-pollinator interactions.

The plant-pollinator network in this system appears to accommodate honey bee interactions, integrating them into the system without major changes, despite honey bees taking a central role (Fig 3). Honey bees may spread pathogens and parasites to some wild bee species via shared flowers [60], or cause other changes to wild plant or pollinator fitness undetectable from looking at changes to flower visitation [18]. Furthermore, we did not detect all occurring plant-pollinator interactions at our transects. Our estimated sampling completeness was 37% overall (see S4 Table), suggesting that we missed many rare interactions. Rare interactions typically involve rare species, which may be particularly sensitive to competition from honey bees [5]. However, understanding honey bee effects on the subset of detected interactions is useful because the most detectable interactions are the most frequent, and are those that contribute most to ecosystem function [61].

Additionally, our study was conducted in only one season, and plant-pollinator interactions are known to be highly variable year to year [62]. This variability is largely because of species and interaction turnover, however, and network structure is expected to be more consistent between years [62], so our conclusions about network structure might be expected to hold in other years. The sampled year 2019 was drier than the 20-year average (S9 Table), meaning floral resources should have been more limiting than in a normal year [63], and therefore we should have been more likely to see competitive effects of honey bees on native pollinators in

2019. We also sampled in 2018, another very dry year (S10 Table), and though the data were not of high enough quality to publish, they indicated the same results.

## Conclusions

Honey bees have been frequently associated with wild pollinator decline, and because regulating honey bee densities might assist wild pollinator conservation, it is worth establishing whether honey bees are a significant threat to wild pollinators. Our study took place in a part of Canada with high bee diversity, in a dry year when floral resources for pollinators should have been more limiting than usual. In this setting we found no effects of honey bees on wild pollinator-plant network metrics, suggesting that even at high densities, honey bees do not detectably cause shifts in wild pollinator foraging habits in this region. Future studies should control for the effects of flower abundance and diversity on plant-pollinator network structure, and distinguish the effects of honey bee interactions on total network structure from effects of honey bees on just the wild pollinator interactions with plants.

## Supporting information

**S1 File.**
(DOCX)

**S1 Fig. Abundance of honey bees caught visiting flowers (full season all taxa dataset), pooled across the full season per transect, with transects ordered by increasing honey bee abundance, and coloured by distance from bee hives.** In the transect names, 100 indicates 100 m, 500 indicates 500 m, and 5000 indicates 5000 m distances from hives.
(TIFF)

**S2 Fig. Visual representation of the spatial autocorrelation of residuals from the model for each response variable for the full season all taxa dataset, for which honey bee abundance was a significant predictor (before Bonferroni-Holm correction) when it was the only predictor in the model, for full season dataset.** Each panel is a map of the transect locations in space (compare to Fig 2). Circles indicate the residual from each transect, with circle size proportional to residual size (smaller circle = better model fit for that transect). Colour indicates the sign of the residual; blue shows values lower than 0 and red values higher than 0. Circles close together in space having the same colour and size would indicate that spatial autocorrelation might be a problem, in which case a special correlation structure would be likely to be selected during model selection.
(TIFF)

**S3 Fig. Total abundance of honey bees caught visiting flowers, with data pooled across all transects.** The entire season was split into three: collection rounds 1–4 represented "early" season (May 28th-July 7th), 5–7 represented "mid" season (July 8th-July 31st), and 8–10 represented "late" season (August 1st-August 28th).
(TIFF)

**S4 Fig. Rarefaction curve for interaction richness.** This figure shows, for a given number of re-sampled interactions from our full season all taxa dataset (x-axis), the mean number of unique interactions detected (y-axis). There was a total observed interaction richness of 654, and Chao1 estimated 1,779 interactions (95% confidence interval: 1500–2148 interactions), meaning 31–44% of the estimated interactions were observed.
(TIFF)

**S1 Table. Longitude and latitude for each hive location and sampling transect, and collection effort at each transect.** The northernmost, central, and southernmost hive locations are listed as Bee48, Bee32, and Bee16 respectively, indicating their number of hives. Each transect is indicated by its treatment (100 m, 500 m, or 5000 m distance from a hive location). Letters indicate each replicated set of distances from hives (See Fig 1). G5000 indicates the new location sampled at once sampling at F5000 was discontinued mid-season.
(DOCX)

**S2 Table. List of references and resources used in species identifications.** Full citations are listed in the References section of the Supporting Information.
(DOCX)

**S3 Table. Pearson correlations between the response variables (network metrics).** Each row and column header refers to the following metrics, respectively: G, generality; V, vulnerability; Poll. NO, pollinator niche overlap; Pl. NO, plant niche overlap; Poll. FC, pollinator functional complementarity; Pl. FC, plant functional complementarity; WNODF, weighted nestedness based on overlap and decreasing fill; M, modularity; WC, weighted connectance; LD, link density; ISA, interaction strength asymmetry; IE, interaction evenness.
(DOCX)

**S4 Table. Pearson correlations between predictor variables.** "Collections" refers to the number of collection rounds at each transect, "Honey bee abundance" refers to honey bee abundance at each transect, "Flower species" refers to the number of flowering species at each transect, and "Flower abundance" refers to the number of individual flowers at each transect.
(DOCX)

**S5 Table. Percent sampling completeness and 95% confidence intervals, calculated as the number of unique interactions observed, divided by the Chao1 estimated total number of unique interactions, and multiplied by 100, for the full season all taxa, mid-season all taxa, and full season bees-only datasets.** The "Lower 95%" column indicates the observed interactions divided by the lowest Chao1 estimated total number of true interactions, and the "Upper 95%" column indicates the observed interactions divided by the highest Chao1 estimated total number of true interactions (which is why the "Lower" values are higher). Shaded cells denote networks that had too few species for some network metrics to be meaningful, so these replicates were excluded from analysis for the metrics: generality, vulnerability, plant and pollinator niche overlap, and nestedness.
(DOCX)

**S6 Table. Identifications of insect pollinators to species-level or morphospecies level.** Morphospecies identifications are listed by "[Genus] sp. #". Some species could not be differentiated between genera, and so both genera are listed along with the epithet "sp". Specimens listed beside "cf" (confer, meaning compare with) are specimens that were damaged or for which taxonomic keys are insufficient, and these were compared to other specimens to determine identification. Numbers of each (morpho)species are given for each distance from hives, despite the fact that we used honey bee abundance, rather than distance from hive as the predictor variable in our analyses.
(DOCX)

**S7 Table. Identifications of flowering species from each distance from honey bee hives to species level.**
(DOCX)

**S8 Table. Network size (number of interactions) for each network that was analyzed from each transect.** Shaded cells denote networks that had too few species for some network metrics to be meaningful, so these replicates were excluded from analysis for the metrics: generality, vulnerability, plant and pollinator niche overlap, and nestedness. Mean network size and range in network size are shown for each dataset.
(DOCX)

**S9 Table. Comparison of the full season meta-network metrics calculated for networks created by pooling data from all transects at each distance from hives, over the whole season.** There was a general increase in each network metric with distance to hives, with the exceptions of plant niche overlap, functional complementarity, and generality. Bolded values indicate results that were unexpected based on the literature and/or expectations if competition between honey bees and wild pollinators is occurring. Positive (+) indicates a positive correlation between honey bee abundance and the metric, while negative (-) indicates a negative correlation, with distance from hives as a proxy for honey bee abundance (even though this was not a perfect proxy, as can be seen in S1 Fig).
(DOCX)

**S10 Table. Historical data obtained from the Alberta Climate Information Service (ACIS) Verger AGCM weather station in Southern Alberta, located on the University of Alberta's Mattheis Research Ranch.**
(DOCX)

## Acknowledgments

We thank Brittany Wingert, Irene Jimenez Roncancio, James Glasier, Greg Pohl, Lincoln Best, and the University of Calgary for their assistance in the identification of specific insect taxa. We also thank assistants Alexandra Johnson and Janelle Goodine, and volunteers Connor Nelson, Olivia DeBourcier, Zachary Roote, Ferf Brownoff, Sara Peterson, Olivia Hrehoruk, Rykkar Jackson, and Victoria Dubord for all their support on this project. We thank Edwin and Ruth Mattheis for donating the Mattheis Research Ranch to the University of Alberta, and Marcel Busz and Lisa Raatz for assistance with logistics at the ranch. We also thank the Eastern Irrigation District for allowing the use of their land. Honey bees were generously supplied by Pankratz Beekeeping. We thank Cameron Carlyle and Mark Poesch for helpful discussion.

## Author Contributions

**Conceptualization:** Sydney H. Worthy, John H. Acorn, Carol M. Frost.

**Data curation:** Sydney H. Worthy, John H. Acorn.

**Formal analysis:** Sydney H. Worthy.

**Funding acquisition:** Sydney H. Worthy, Carol M. Frost.

**Investigation:** Sydney H. Worthy, John H. Acorn, Carol M. Frost.

**Methodology:** Sydney H. Worthy, John H. Acorn, Carol M. Frost.

**Project administration:** Sydney H. Worthy.

**Resources:** Carol M. Frost.

**Supervision:** John H. Acorn, Carol M. Frost.

**Visualization:** Sydney H. Worthy.

**Writing – original draft:** Sydney H. Worthy.

**Writing – review & editing:** Sydney H. Worthy, John H. Acorn, Carol M. Frost.

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
