## [Decision Letter · Decision Letter 0]

17 Apr 2023

PONE-D-23-06336Honey bees (Apis mellifera) modify plant-pollinator network structure, but do not alter wild species’ interactionsPLOS ONE

Dear Dr. Worthy,

Thank you for submitting your manuscript to PLOS ONE. After careful consideration, we feel that it has merit but does not fully meet PLOS ONE’s publication criteria as it currently stands. Therefore, we invite you to submit a revised version of the manuscript that addresses the points raised during the review process.

We look forward to receiving your revised manuscript.

Kind regards,

Ramzi Mansour

Academic Editor

PLOS ONE

Journal Requirements:

4. We note that Figure 2 in your submission contain [map/satellite] images which may be copyrighted. All PLOS content is published under the Creative Commons Attribution License (CC BY 4.0), which means that the manuscript, images, and Supporting Information files will be freely available online, and any third party is permitted to access, download, copy, distribute, and use these materials in any way, even commercially, with proper attribution. For these reasons, we cannot publish previously copyrighted maps or satellite images created using proprietary data, such as Google software (Google Maps, Street View, and Earth). For more information, see our copyright guidelines: http://journals.plos.org/plosone/s/licenses-and-copyright.

Reviewers' comments:

Reviewer's Responses to Questions

**Comments to the Author**

1. Is the manuscript technically sound, and do the data support the conclusions?

Reviewer #1: Yes

Reviewer #2: Yes

2. Has the statistical analysis been performed appropriately and rigorously? 

Reviewer #1: Yes

Reviewer #2: Yes

3. Have the authors made all data underlying the findings in their manuscript fully available?

Reviewer #1: Yes

Reviewer #2: Yes

4. Is the manuscript presented in an intelligible fashion and written in standard English?

Reviewer #1: Yes

Reviewer #2: Yes

5. Review Comments to the Author

Reviewer #1: 

The paper is extremely interesting and very well written. It deals with a truly debated topic without starting from preconceptions but based only on knowledge of the best literature on this topic. I would suggest to the authors to include in the article, especially in the introduction, a better description of the diversity between Apis mellifera and other Apoidea and pollinators. It would perhaps be useful to further clarify that Apis mellifera is a species introduced by man in Canada (when?) and to give some simple data on the numerical and geographical consistency of beekeeping in Canada and in the region considered. This could be useful in the discussion but above all it clarifies some fundamental concepts to non-apidologist readers. I tried to give some hints in the notes. As far as the materials and methods are concerned, it would be important to clarify whether the area being tested was subject to beekeeping already before the research or whether the apiaries were set up in an area hitherto free from beekeeping. it would have been interesting to have a year or a period without honey bees, but the robustness of the data and analyzes is not affected. The results are very interesting and well contextualised.

Reviewer #2: 

Title: Honey bees (Apis mellifera) modify plant-pollinator network structure, but do not alter wild species’ interactions

Authors: Sydney H. Worthy, John H. Acorn, Carol M. Frost

Brief summary: The paper deals with the experimental introduction of honey bees to a Canadian grassland and evaluation of plant-pollinator interactions at varying distances from the introduced hives with the aim of studying whether honey bees alter wild pollinator interactions, or if they merely alter total network structure by adding their own interactions. The authors found that honey bees increased the network metrics of pollinator and plant functional complementarity and decreased interaction evenness. However, in networks constructed from just wild pollinator interactions, honey bee abundance did not affect any of the metrics calculated. Thus, all network structural changes to the full network (including honey bee interactions) were due only to honey bee-plant interactions, and not to honey bees causing changes in wild pollinator-plant interactions. The Authors, on the basis of their results suggest that honey bees did not alter wild pollinator foraging patterns in this system, even in a year that was drier than the 20-year average.

Points of strength and weakness of the paper: In general, the study seems to be well conducted, however, the description of the methodology needs to be shortened and the results expanded by using some of the supplementary material, for example, regarding pollinator species and the visited plants.

Specific comments

Page 10, Line 220: change “complemenarity” with “complementarity”

Page 13, Line 287: “… it would suggest that that response ….”: “Please, change as follows “it would suggest that response …”

Page 16, Lines 336-340: “Bees (including honey bees) accounted for 872 (48.1%) of the total interactions. Of the remaining interactions, flies accounted for 494 (27.2%), butterflies accounted for 130 (7.2%), ants accounted for 104 (5.7%), beetles accounted for 100 (5.5%), wasps accounted for 62 (3.4%), true bugs accounted for 27 (1.5%), and day-flying moths accounted for 25 (1.4%).”: Please, change with “Bees (including honey bees) accounted for 872 (48.1%) of the total interactions. Of the remaining interactions, flies accounted for 494 (27.2%), butterflies for 130 (7.2%), ants for 104 (5.7%), beetles for 100 (5.5%), wasps for 62 (3.4%), true bugs for 27 (1.5%), and day-flying moths for 25 (1.4%). “

Page 23, Line 460: “However, (17) did …”: Please, change in “However, Magrach et al (17) did …”

Page 23, Line 479: “…(Worthy et al in review).”: Please, remove or add the reference.

Page 25, Line 498: “In their network study, Bendel et al (52) determined …” instead of “In their network study, (52) determined …”

6. PLOS authors have the option to publish the peer review history of their article (what does this mean?). If published, this will include your full peer review and any attached files.

Reviewer #1: **Yes: **Paolo Fontana

Reviewer #2: No

---

## [Author Response · Author response to Decision Letter 0]

25 May 2023

Dear Dr. Bendaña and Dr. Mansour,

Thank you for allowing us the opportunity to re-submit our manuscript to PLOS ONE. We have revised our manuscript “Honey bees (Apis mellifera) modify plant-pollinator network structure, but do not alter wild species’ interactions” in light of the reviewers’ comments, and we believe that the manuscript is improved as a result. We are grateful for these comments and have responded specifically to each suggestion. We have marked our responses with ‘>’. 

Additionally, we have addressed the journal requirements listed in the second decision letter, including re-making Figure 2. We stated the Open Government License used for map files, and also added a full ethics statement in the Methods section. 

We thank you for your time and consideration.

Yours sincerely,

Sydney Worthy

John Acorn

Carol Frost

Reviewer #1 (Comments to Author): 

The paper is extremely interesting and very well written. It deals with a truly debated topic without starting from preconceptions but based only on knowledge of the best literature on this topic.

> We thank the reviewer for these positive comments.

 I would suggest to the authors to include in the article, especially in the introduction, a better description of the diversity between Apis mellifera and other Apoidea and pollinators. 

>We added “Wild insect pollinators include many species, including other bees, wasps, and many other orders, including flies, butterflies, moths and beetles, which are diverse in size, sociality, and floral trait preferences” in the Introduction, to address this comment and better describe the diversity of insects that “wild pollinators” includes, highlighting that it includes but is not limited to other Apoidea (L48-50).

We also quantify this diversity of wild pollinators in the first paragraph of the results section (281 pollinator species; L345).

It would perhaps be useful to further clarify that Apis mellifera is a species introduced by man in Canada (when?) and to give some simple data on the numerical and geographical consistency of beekeeping in Canada and in the region considered. This could be useful in the discussion but above all it clarifies some fundamental concepts to non-apidologist readers. I tried to give some hints in the notes. 

> We have addressed this comment by including a line about Apis mellifera being native to Europe, Africa, and the Middle East in the introduction (L43-44). Regarding consistency of beekeeping in the region, we think that Canada is too large of a region to be relevant, as there is high wild pollinator species composition change from east to west and south to north in Canada. However, to address the request for regional beekeeping information we have now added to the methods section “Honey bees have been present in southern Alberta for over a century, with hive numbers in Alberta increasing by about 4000 hives per year over the past 40 years. Though sporadic bee keeping has occurred in and around the study area in the past, the Eastern Irrigation District, owner of the land surrounding the Mattheis Research Ranch, reported no known apiaries within 19 km of the ranch’s boundary line within the study year or the previous year. Apiaries are required by law to register their hive(s) annually to the Provincial Apiculturist” (lines 175-179).

As far as the materials and methods are concerned, it would be important to clarify whether the area being tested was subject to beekeeping already before the research or whether the apiaries were set up in an area hitherto free from beekeeping. 

>We have now clarified that “sporadic bee keeping has occurred in and around the study area in the past” (L175-176), but that “the Eastern Irrigation District, owner of the land surrounding the Mattheis Research Ranch, reported no known apiaries within 19 km of the ranch’s boundary line within the study year or the previous year”.

Reviewer 1’s notes in the marked up manuscript:

Specify where Apis mellifera is autochthonous (Europe, Africa, Middle East..) and that it was then introduced by man in other areas (N, S America, Oceania, Asia)

>We have added that the native range of Apis mellifera is “in Europe, Africa, and the Middle East” (L43-44)

It must be taken into account that the maximum food consumption, especially of pollen, is before the peak of the adult bee population, because larvae need the maximum food effort. In this phase prior to the population peak, the key resource is pollen, while subsequently nectar is strategic

>We have added a line to address this (L145-146)

What is not clear to me, is if the 3 apiaries where placed in their site also before trials (there was a beekeeping activity before the research or not) and if sempling of Non-Apis pollinators have been done before honey bees displacement.

>See response to the same comment above, which we have addressed (L173-179)

When? (In reference to hive placement)

>We have added “In 2019” (L168)

Better explain the distribution of transects

>We have added “The 100 m and 500 m distances were chosen to achieve high and medium honey bee densities. We chose 5000 m as the minimum distance for sites intended to be without honey bees, because on average they do not tend to travel farther than this distance” (L182-185). 

Model? (In reference to the model of dissecting microscope used)

>Microscope models are now listed (L211)

Specify the collection (In reference to where the reference specimens were housed that we used to confirm insect IDs)

>We have expanded this to read “and comparison with reference specimens from the University of Alberta Strickland Museum and University of Calgary Museum of Zoology entomological collections” (L212-214).

Honey bees can fly up to 13-14 km to collect pollen

>We acknowledge that this is possible. However, we don’t feel that adding this statement is necessary, in the interests of Reviewer 2’s request to shorten the methods, because we are reporting that honey bees arrived at that transect, so clearly it was possible.

Reviewer #2: 

Points of strength and weakness of the paper: In general, the study seems to be well conducted, however, the description of the methodology needs to be shortened and the results expanded by using some of the supplementary material, for example, regarding pollinator species and the visited plants.

> We added L L173-179 and L182-185 to the methods to address Reviewer 1’s comments, and added the additions mentioned below in response to Reviewer 2’s requests for additional detail in the methods. However, we have also shortened the methods by removing a few lines where possible (see tracked changes version). We are worried about removing any more detail because the analysis is complicated, and we feel that we have already made it as concise as possible for it to be comprehensible. 

We have added Figure S5 from the Supporting Information to the main text, which shows the pollinator species and the visited plants. We have made this Figure 3 and have changed the previous Figure 3 to Figure 4.

Specific comments:

Page 10, Line 220: change “complemenarity” with “complementarity”

> Done (L238)

Page 13, Line 287: “… it would suggest that that response ….”: “Please, change as follows “it would suggest that response …”

> Done (L304)

Page 16, Lines 336-340: “Bees (including honey bees) accounted for 872 (48.1%) of the total interactions. Of the remaining interactions, flies accounted for 494 (27.2%), butterflies accounted for 130 (7.2%), ants accounted for 104 (5.7%), beetles accounted for 100 (5.5%), wasps accounted for 62 (3.4%), true bugs accounted for 27 (1.5%), and day-flying moths accounted for 25 (1.4%).”: Please, change with “Bees (including honey bees) accounted for 872 (48.1%) of the total interactions. Of the remaining interactions, flies accounted for 494 (27.2%), butterflies for 130 (7.2%), ants for 104 (5.7%), beetles for 100 (5.5%), wasps for 62 (3.4%), true bugs for 27 (1.5%), and day-flying moths for 25 (1.4%).”

> Done (L353-356)

Page 23, Line 460: “However, (17) did …”: Please, change in “However, Magrach et al (17) did …”

> Done (L489)

Page 23, Line 479: “…(Worthy et al in review).”: Please, remove or add the reference.

> The reference was removed (L489)

Page 25, Line 498: “In their network study, Bendel et al (52) determined …” instead of “In their network study, (52) determined …”

 > Done (L509)

---

## [Editor Report · Decision Letter 1]

4 Jun 2023

Honey bees (Apis mellifera) modify plant-pollinator network structure, but do not alter wild species’ interactions

PONE-D-23-06336R1

Dear Dr. Worthy,

We’re pleased to inform you that your manuscript has been judged scientifically suitable for publication and will be formally accepted for publication once it meets all outstanding technical requirements.

Kind regards,

Ramzi Mansour

Academic Editor

PLOS ONE
---

## [Editor Report · Acceptance letter]

3 Jul 2023

PONE-D-23-06336R1 

Honey bees (*Apis mellifera*) modify plant-pollinator network structure, but do not alter wild species’ interactions 

Dear Dr. Worthy:

I'm pleased to inform you that your manuscript has been deemed suitable for publication in PLOS ONE. Congratulations! Your manuscript is now with our production department. 

Kind regards, 

on behalf of

Dr. Ramzi Mansour 

Academic Editor

PLOS ONE